# Genome-Wide Analysis of LncRNA in Bovine Mammary Epithelial Cell Injuries Induced by Escherichia Coli and Staphylococcus Aureus

**DOI:** 10.3390/ijms22189719

**Published:** 2021-09-08

**Authors:** Changjie Lin, Yifan Zhu, Zhiyu Hao, Haojun Xu, Ting Li, Jinghan Yang, Xi Chen, Yingyu Chen, Aizhen Guo, Changmin Hu

**Affiliations:** 1Department of Clinical Veterinary Medicine, Faculty of Veterinary Medicine, Huazhong Agricultural University, Wuhan 430070, China; 875155626@webmail.hzau.edu.cn (C.L.); a845911819@163.com (H.X.); tasjlt@webmail.hzau.edu.cn (T.L.); yjh980718@163.com (J.Y.); 2State Key Laboratory of Agricultural Microbiology, Huazhong Agricultural University, Wuhan 430070, China; avander1@163.com (Y.Z.); hzy0908@webmail.hzau.edu.cn (Z.H.); chenxi@mail.hzau.edu.cn (X.C.); chenyingyu@mail.hzau.edu.cn (Y.C.); aizhen@mail.hzau.edu.cn (A.G.); 3Department of Preventive Veterinary Medicine, Faculty of Veterinary Medicine, Huazhong Agricultural University, Wuhan 430070, China

**Keywords:** mastitis, LncRNA, *E. coli*, *S. aureus*, lncRNA–miRNA–mRNA networks

## Abstract

*Escherichia coli* and *Staphylococcus aureus* are two common pathogenic microorganisms that cause mastitis in dairy cows. They can cause clinical mastitis and subclinical mastitis. In recent studies, lncRNAs have been found to play an important role in the immune responses triggered by microbial inducers. However, the actions of lncRNAs in bovine mastitis remain unclear. The purpose of this study was to investigate the effects of bovine mammary epithelial cell injuries induced by treatment with *E. coli* and *S. aureus**,* and to explore the lncRNA profile on cell injuries. The lncRNA transcriptome analysis showed a total of 2597 lncRNAs. There were 2234 lncRNAs differentially expressed in the *E. coli* group and 2334 in the *S. aureus* group. Moreover, we found that the *E. coli* and *S. aureus* groups of maternal genes targeted signaling pathways with similar functions according to KEGG and GO analyses. Two lncRNA–miRNA–mRNA interaction networks were constructed in order to predict the potential molecular mechanisms of regulation in the cell injuries. We believe that this is the first report demonstrating the dysregulation of lncRNAs in cells upon *E. coli* and *S. aureus* infections, suggesting that they have the potential to become important diagnostic markers and to provide novel insights into controlling and preventing mastitis.

## 1. Introduction

Mastitis is one of the most widespread diseases in bovine populations, seriously endangering the health of dairy cows and the development of the dairy-farming industry. It is usually caused by physical and chemical stimulation or induced by pathogenic microorganisms [1,2]. *Escherichia coli* (*E. coli*) and *Staphylococcus aureus* (*S. aureus*) are the common pathogenic bacteria that lead to intramammary infections [3,4]. Zadoks et al. found that different pathogenic microorganisms had different effects on mastitis [5]. The mastitis induced by *E. coli* was acute, with severe clinical symptoms, and the animals could be cured by their own immunity or antibiotic therapy [6]. In contrast, mastitis caused by *S. aureus* was often chronic or recessive, which made it difficult to diagnose [7,8,9]. The evidence suggested that there was a difference in pathogenesis between *E. coli* and *S. aureus*. Therefore, both bacteria were studied in vitro for their roles in mastitis.

Bovine mammary epithelial cells play a significant role in the first line of immune defense in bovines. They can secrete a variety of bioactive substances, such as inflammatory factors, chemokines, β defensins, and interferon γ (IFNγ), to resist the invasion of pathogenic microorganisms [10]. It has been proven that the immune responses of bovine mammary epithelial cells to *E. coli* and *S. aureus* infections differ [11,12,13]. In addition, researchers have also identified an obvious difference in the epithelial immune responses in milk [14] and mammary cells [15]. Emerging studies have reported that bovine mastitis is associated with cell injuries, such as those induced by inflammatory reactions, oxidative stress, and apoptosis [16,17]. Furthermore, cell injuries have been shown to affect lactation ability [18]. Therefore, the exploration of bovine mammary epithelial cell injuries could help in resolving bovine mastitis.

Long non-coding RNAs (lncRNAs) mainly refer to a type of RNA transcript without obvious protein-coding capacity in an organism. Most lncRNAs are transcribed by RNA polymerase II, are capped at their 5′ ends and contain polyadenylated tails at their 3′ ends, similar to mRNAs. Mature lncRNAs are formed through splicing and are usually longer than 200 nucleotides [19,20]. At present, lncRNAs can be detected in biological tissues, organs, pathological samples, and cells cultured in vitro. They are highly evolutionarily conserved [21]. Another important feature of lncRNAs is that they are tissue specific [22]. LncRNAs play a fundamental role at all levels of gene control, including in epigenetic mechanisms and nuclear tissue, as well as RNA processing, stability, and translation. They also participate in the regulation of the physiology and pathology of many diseases [23]. LncRNAs can promote inflammation by promoting the transcription of certain genes [24]. Sponge function is one of the most important regulatory functions of lncRNAs. It has been found that lncRNAs can sequester microRNAs (miRNAs) to exercise the function of competing endogenous RNAs (ceRNAs) and participate in the regulation of cell proliferation, metastasis, inflammation, autophagy and apoptosis [25]. However, there are few studies on the involvement of lncRNAs in the regulation of bovine mastitis.

In this study, we constructed a model of mastitis in vitro and explored the injuries to bovine mammary epithelial cells induced by *E. coli* and *S. aureus*. High-throughput sequencing technology was used to profile the expression of lncRNAs in *E. coli*- and *S. aureus*-induced and healthy bovine mammary epithelial cells and analyze the basic characteristics and differential expression. Furthermore, we also predicted the miRNAs adsorbed by the lncRNAs that were significantly differentially expressed in the two bacterial inducers and constructed the ceRNA regulatory network of LncRNA–miRNA–mRNA therein. This lays the foundation for an in-depth exploration of the function of lncRNA regulation in mastitis diseases, analyses of the pathogenesis of mastitis in dairy cows and the screening of effective therapeutic drugs.

## 2. Results

### 2.1. Cell Injuries Induced by Inactivated E. coli and S. aureus

In order to confirm the damaging effects of *E. coli* and *S. aureus* in MAC-T cells, we used them to treat cells for 24 h and measured the percentages of apoptotic cells, cell cycle progression, and the levels of ROS. The mRNA levels for pro-inflammatory cytokines (such as IL-1β, IL-6 and TNF-α) were analyzed by qPCR (Figure 1A). We observed that there were significant increases of IL-1β, IL-6, and TNF-α in the *E. coli* and *S. aureus* as compared with the control group (*p* < 0.001). The cell viability of MAC-T (CCK8 test) cells after 24 h of stimulation by *E. coli* and *S. aureus* was used to determine the toxic effects of *E. coli* and *S. aureus* on cells (Figure 1B). The damaging effects of *E. coli* as well as *S. aureus* were significant.

Compared with those in the control group, the intracellular reactive oxygen levels in the *E. coli*- and *S. aureus*-stimulated groups were significantly increased (Figure 1C–F). Compared with the control group (Figure 1G), the *E. coli*-stimulation group (Figure 1I) and the *S. aureus* group (Figure 1H) showed significantly increased rates of apoptosis of the cells. *E. coli* and *S. aureus* promoted cell-cycle arrest in the G0/G1 phase (Figure 1K–M) and reduced the proportions of cells in the G2 and S phases to varying degrees, thereby inhibiting cell proliferation.

### 2.2. Identification of lncRNAs in Bovine Mammary Cells Injured by Inactivated E. coli and S. aureus

Since the critical functions of lncRNAs in various diseases are not well understood, we were interested in investigating the levels of lncRNAs in bovine mammary cells. Therefore, we carried out high-throughput sequencing to explore gene regulation by inactivated *E. coli* and *S. aureus*. Totals of 81,512,268, 64,310,018, and 82,433,054 raw reads were obtained from the *E. coli* group, *S. aureus* group and control group, respectively (Appendix A). Through in silico analyses via different bioinformatic approaches, including using the Cutadapt software, for data filtering and quality control, 81,506,016, 64,304,914, and 82,429,030 high-quality reads were retained, respectively, and the proportions of net reads were between 99.990% and 99.992%. A total of 2597 lncRNAs, including 680 intergenic lncRNAs, 998 intronic antisense lncRNAs, 565 natural antisense lncRNAs, and 354 other lncRNAs, were identified (Figure 2B). There were 2225 mutual lncRNAs, whereas 89 and 119 lncRNAs were exclusively regulated upon *E. coli* or *S. aureus* transfection, respectively (Figure 2A, Appendix A). The lengths of our lncRNAs mainly varied from 200 base pairs (bp) to 5000 bp (Figure 2C). At the same time, we found that the lncRNA expression was low according to the sequencing results, with around three counts (Figure 2C) but also that there were high count numbers, possibly because lncRNAs are more stable than other lncRNAs.

### 2.3. Profiles of lncRNAs Differentially Expressed upon Injury of Bovine Mammary Cell by Inactivated E. coli and S. aureus

In order to identify the lncRNAs associated with cell injuries, the lncRNA expression in the *E. coli* group, the *S. aureus* group, and the control group was analyzed (Appendix A). The differentially expressed lncRNAs of the two treated groups (compared with the control group) were used for cluster analysis (Figure 3A,B). By constructing a heat map, we were able to show that the two inactivated-bacterium-treated groups produced two separate clusters, indicating that the expression patterns of lncRNAs in the two inactivated-bacterium-treated groups were different from those for the control group, and that the observed differences were significant (Figure 3C,D). At the same time, 76 significantly upregulated and 162 significantly downregulated lncRNAs were detected in the *E. coli* group, and 80 significantly upregulated and 188 significantly downregulated lncRNAs were detected in the *S. aureus* group (Figure 2D, Appendix A).

### 2.4. Comparison of lncRNA Characteristics in Bovine Mammary Cell Injuries by Inactivated E. coli and S. aureus

Recent studies have shown that lncRNAs can affect the expression of adjacent upstream and downstream genes. Therefore, in order to study the important role that lncRNAs play in the bovine mammary epithelial cell injuries induced by *E. coli* and *S. aureus*, the target genes of differentially expressed lncRNAs were analyzed by GO and KEGG enrichment analysis (Appendix A). The functional annotation by GO indicated that the predicted target genes in the *E. coli* group were related to catalytic activity, arginine metabolic processes, methylated histone binding, as well as mitochondrion and palmitoyltransferase activity (Figure 4B, Appendix A). In the KEGG pathway analysis, we found that the lncRNA target genes in the *E. coli* group were enriched in cell-injury-related pathways, such as tight junctions, ErbB signaling pathways, biosynthesis of amino acids, and Fc epsilon RI signaling pathways (Figure 5A, Appendix A). The GO analysis of differentially expressed lncRNAs in the *S. aureus* group showed that the meaningful terms were related to the regulation of epithelium migration, regulation of signal transduction, cell junction, RNA polymerase II, core complex, transcription regulator activity, and amino acid binding (Figure 4A, Appendix A). The KEGG pathway analysis revealed that lncRNA target genes were enriched in the PI3K–Akt signaling pathway, focal adhesion, cell cycle, and bacterial invasion of epithelial cells (Figure 5B, Appendix A). From these analyses, we speculated that lncRNAs may act through the cis-regulation of protein-coding genes to regulate the immune reaction and the growth metabolism in the two different stimulation groups.

### 2.5. Target Predictions: LncRNA–miRNA–mRNA Regulatory Network Analysis

Previous studies have suggested that lncRNAs could regulate the expression of target genes by acting as a molecular sponge to sequester miRNA. In this study, the potential miRNA targets of all the differentially expressed lncRNAs were predicted based on complementary base pairing. We constructed an lncRNA–miRNA–mRNA network by merging lncRNA–miRNA regulatory networks and miRNA–mRNA networks, including 29 mRNAs, eight lncRNAs, and 11 miRNAs in the *E. coli* group (Figure 6A) and 29 mRNAs, nine lncRNAs, and 11 miRNAs in the *S. aureus* group (Figure 6B, Appendix A).

In the network of the *S. aureus* group, many cell-injury-related miRNAs were observed. miR-346, miR-370, miR-484, and miR-149-3p were shown to inhibit proliferation or induce apoptosis and inflammation by suppressing the expression of the target genes. The results from the present study also suggest that some miRNAs might interact with multiple lncRNAs. For instance, miR-149-3p might interact with LOC112443417, LOC112441776, LOC104971359, and LOC100848507. These lncRNAs are thought to play an important role in *E. coli*- and *S. aureus*-induced injury.

Similarly, in the group treated with inactivated *E. coli*, we predicted many mRNAs related to inflammation, such as MAPK12, RELA, MAPK3, PIK3R6, and JAK3. Additionally, multiple miRNAs can be predicted to interact with them. For example, p38 can be regulated by miR-2454-5p, miR-2305, miR-1777b, miR-2442, and miR-149-3p. Interestingly, we found that the same miRNAs were predicted in the two different stimulus groups, suggesting that these lncRNAs might play similar roles in mastitis.

### 2.6. Differential Expression of lncRNA Was Verified by Quantitative PCR

We randomly selected 10 lncRNAs to verify the lncRNA-seq results. It can be seen from Figure 7 that the qPCR results were consistent with the data obtained from the RNA-seq results, showing that our sequence data were reliable.

## 3. Discussion

Bovine mastitis is a common disease that can cause serious damage to public health, animal welfare and the global economy [26,27]. The treatment of mastitis still involves antibiotics, which can easily promote drug resistance in pathogenic microorganisms, leading to an increase in drug-resistant strains and unhealthy foods [28,29,30]. Therefore, it is urgent that we explore new and effective drugs that can deliver targeted treatment for bovine mammary injury [31,32]. Recent experiments have shown that lncRNAs can play an important role in various diseases [33,34,35], and studies have shown that lncRNA-targeted drugs can be used to treat diseases [36]. The characteristic expression of lncRNAs in bovine mastitis and their mechanism of action are not fully understood. In the present study, we used inactivated *E. coli* and inactivated *S. aureus* to treat MAC-T cells, to simulate the occurrence and development of mastitis. The levels of IL-1β, IL-6, and TNF-α were significantly higher with treatment, indicating that the cell model of mastitis had been successfully established [37,38,39]. To facilitate the further identification of the mechanisms of mastitis, a comprehensive assessment of this mastitis cell model was conducted, using a CCK8 assay and flow cytometry. In addition, when compared to that in the control group, the cell viability in the treated groups was inhibited, the ROS levels and apoptosis rate increased, and cell-cycle arrest was observed in both the *E. coli* and *S. aureus* groups. The mastitis cell model was therefore optimized. Therefore, we were able to perform high-throughput sequencing analysis of lncRNAs from treated and untreated cells and study the differential expression of lncRNAs in bovine mastitis more comprehensively. Our sequencing results not only provide comprehensive data on the expression profile of lncRNAs in bovine mastitis, which greatly enrich the transcript genome database for cattle, they also provide a theoretical basis for finding molecular targets that could be used to diagnose and treat different kinds of mastitis (Figure 8).

In this study, we found 2342 differentially expressed lncRNAs in the *E. coli* group and 2313 differentially expressed lncRNAs in the *S. aureus* group. A total of 2225 lncRNAs were differentially expressed in the two different stimulation groups. We chose 10 lncRNAs for verification. The qRT-PCR analysis showed that the upregulated and downregulated lncRNAs were consistent with the RNA-seq data. The results indicate that these differentially expressed genes may play a vital role in the occurrence of mastitis. LncRNAs are upstream regulators of mRNAs, and we concluded that inactivated *E. coli* and inactivated *S. aureus* have a high degree of similarity in regulating lncRNAs. Therefore, we could find potential lncRNAs as targets for the diagnosis or treatment of mastitis. Furthermore, the current findings may provide new insights of relevance for mastitis drug development.

The functions of lncRNAs have not been fully elucidated. They can interact with coding genes in both cis and trans manners to fulfill their functions [40]. It was found that a new lncRNA, LRRC75A-AS1, could protect its maternal gene (*LRRC75A*) from being degraded and promote inflammation [41]. Therefore, we performed KEGG and GO analyses to predict the functions of lncRNAs based on the functions of the maternal genes. We found that the maternal genes of lncRNAs differentially expressed upon treatment with inactivated *E. coli* and inactivated *S. aureus* enriched in functions are highly consistent. GO analysis showed that more maternal genes from the two groups of differentially expressed lncRNAs were involved in the RNA transcription process, as well as functionally related effects such as signal transduction. The upregulated lncRNAs were mainly involved in the establishment of mitochondrion localization, the mitochondrion Prp19 complex and the kinesin complex. The downregulated lncRNAs were mainly involved in epidermal growth factor receptor binding and epithelium migration, as an extrinsic component of membranes and as part of the cell–cell junction. KEGG analysis allowed us to better understand the complex network in the stimulation process. Our results show that the target genes of DE lncRNAs are highly enriched in focal adhesion, the adherens junction, the biosynthesis of amino acids, the cell cycle, and the PI3K–Akt signaling pathway. The signaling pathways revealed by the enrichment of the lncRNA maternal genes had a high degree of correlation with the related phenotypes of the cells in the mastitis model that we verified. For example, under inflammatory conditions, a damaged tight-junction barrier can increase paracellular permeability and cause cell damage, such as erosion, ulcers, and apoptosis. It can also stimulate the release of pro-inflammatory cytokines, such as TNF-α and IFN-γ, ultimately worsening the inflammation [42,43]. It has been reported that cell-cycle arrest can also cause cell apoptosis in response to DNA damage [44]. The PI3K–Akt signaling pathway has often been reported to regulate the biological processes of inflammation, apoptosis, and cell proliferation [45,46,47]. These ideas were supported by our findings, and lncRNAs may play an important role in cell damage through cis functions. These possible associations should be further explored in future research.

Salmena et al. offered a hypothesis regarding competitive endogenous RNAs (ceRNAs), suggesting that lncRNAs, mRNAs and other types of RNA could act as natural miRNA sponges through pairing with microRNA response elements (MRE). The separation of miRNAs and mRNAs hinders the function of miRNAs [48,49]. Many studies have confirmed this hypothesis. LncRNAs affect the occurrence of mastitis and other diseases through the action of ceRNAs [50]. In our study, an lncRNA–miRNA–mRNA interaction network diagram was constructed based on the principle of ceRNA action, and we found that 18 lncRNAs and 54 mRNAs had a total of 18 miRNA binding sites.

Based on the interaction network diagram, we found that LOC100140121 and LOC104971359 in the *E. coli* group and LOC112442703 and LOC104971369 in the *S. aureus* group could all act as sponges for miR-149-3p. Previous studies have proven that miR-149-3p can participate in the regulation of cell proliferation, apoptosis, and inflammation [51,52]. In our research, miR-149-3p was also predicted to have binding sites in MAPK3, MAPK14, PIK3R2, PRKAG3 and other mRNAs. The MAPK signaling pathway is closely related to cell injury [53]. For example, MAPK3, also known as extracellular-regulated protein kinase 1 (ERK1), is reported to downregulate the expression of miR-483-5p post-transcriptionally, leading to apoptosis and oxidative stress in human cardiomyocytes under hypoxia [54]. Another study reported that inhibiting miR-124 can cause the overexpression of MAPK14, which, in turn, activates the MAPK signaling pathway and promotes lung cell apoptosis and inflammation [55]. In addition, existing research has shown that PIK3R2 can be used as a target for miRNAs [56]. After being regulated by miRNAs, it can affect cell inflammation and apoptosis by changing the activity of the PI3K/AKT signaling pathway. These previous findings are in agreement with our prediction that lncRNAs may participate in the process of cell damage through sponge function. Similarly, the predicted miR-1777b could also be regulated by LOC104976293 and LOC104971359 in the *E. coli* group and LOC107132431 and LOC104971369 in the *S. aureus* group. Its target genes include RELA, NOTCH2, and JAK3. It has been reported that vascular endothelial cells can inhibit the expression of miR-141-3p in a hypoxic environment and promote cell apoptosis by increasing the content of NOTCH2 by targeting it [57]. Previous studies also showed that JAK3 is a key member of the TLR4 signaling pathway, which plays an important role in the occurrence of inflammation [58]. Some studies have reported that miR-221-3p can inhibit its expression by targeting JAK3 in the 3′-UTR region, reduce the expression of IL-10, and promote the secretion of pro-inflammatory cytokines [59]. Moreover, most of the lncRNA and miRNA targets predicted in this study have not yet been studied, so we predicted their functions through the targeted mRNAs. As discussed, LOC104971359 in the *E. coli* group and LOC104971369 in the *S. aureus* group could jointly regulate multiple miRNAs, and the targets predicted by miRNAs are also significantly related to processes related to cell injury, such as inflammation, apoptosis, and cell proliferation, which we verified in this study. We further speculate that lncRNAs may play an important role in mastitis and that their co-targeted miRNAs could also be used as new targets for treating mastitis.

This study did have some limitations. For instance, endoplasmic reticulum injury and the mitochondrial membrane need to be studied in the future. Additionally, more experiments, e.g., using animal models, are needed to verify the functions of these differentially expressed lncRNAs.

## 4. Materials and Methods

### 4.1. Cell and Bacteria

MAC-T cells (an immortalized cow mammary epithelial cell line) were kindly donated by Professor Mark Hanigan of Virginia Tech University. The MAC-T cells were cultured in DME/F12 medium supplemented with 10% fetal bovine serum (Gibco, Waltham, MA, USA) and cultured in a 37 °C, 5% CO_2_ incubator. The cells were digested using 0.25% trypsin and 0.02% EDTA.

*E. coli* (ATCC 25922) and *S. aureus* (ATCC 29213) were donated by Associate Professor Wang Xiangru and Professor Zhou Rui of Huazhong Agricultural University. The *E. coli* was resuscitated in LB solid medium, and the *S. aureus* was resuscitated in TSA medium. After 12 h at 37 °C, a single colony was observed. A single *E. coli* colony was transferred to a bacterial culture bottle containing 10 mL of LB broth, which was placed in a shaker at 220 RMP/min for 12 h. Similarly, a single colony of *S. aureus* was grown in 10 mL of TSB broth using the same method. The bacteria were diluted by factors of 10^5^, 10^6^, and 10^7^, and evenly spread on a solid medium. Single colonies were cultured at 37 °C for 12 h and then counted.

### 4.2. Construction of Mastitis Model in MAC-T Cells Induced by E. coli and S. aureus

According to the above method, the two bacterial strains were diluted to reasonable multiples. The two bacteria were then heated at 63 °C for 30 min. After the MAC-T cells were cultured for 12 h in 6-well plates with 2 × 10^5^ cells/well, the inactive bacterial strains were added at a 10:1 ratio (bacteria:cells). The inactivated bacteria and cells were cocultured for 24 h, and the culture medium was discarded. The cells were washed 2–3 times with cold PBS, and 1 mL of TRIzol (Invitrogen, Carlsbad, CA, USA) was added to each well for RNA extraction. Each treatment was performed in triplicate.

### 4.3. Quantitative Real-Time PCR

Total RNA was harvested using TRIzol and reverse transcribed into cDNA using the Hiscript III Reverse Transcriptase (Vazyme, Nanjing, China). Next, we used the AceQ qPCR SYBR Green Master Mix (Vazyme) and the ViiA™ 7 Real-Time PCR System (Foster city, CA, USA) to determine the relative mRNA levels. Finally, the 2^−ΔΔCt^ method was used for comparative analysis. Table 1 lists the primers used in this study.

### 4.4. Determination of Cell Cycle

A total of 2 × 10^5^ cells/well were seeded in 6-well plates. The cells were digested using trypsin for 3 min before the samples were centrifuged to collect the cells. After washing it 3 times with PBS, the sample was resuspended in 70% ethanol. Next, the cells were fixed at 4 °C for 12 h in the 70% ethanol, washed with PBS again, centrifuged at 1000 rpm for 5 min, and stained with propidium for 30 min in a dark environment (Beyotime, Shanghai, China). Then, the cell cycle was detected using a flow cytometer.

### 4.5. Annexin V-FITC and PI Assay for Apoptosis

A total of 2 × 10^5^ cells/well were seeded in 6-well plates. Cell apoptosis was assayed using the Annexin V-FITC/PI Apoptosis Detection Kit (Vazyme). First, we removed the medium and washed the cells with PBS. After detaching and separating the cells using trypsin for 3 min, the samples were centrifuged to collect the cells. After washing it 3 times with PBS, the sample was resuspended in 100 µL of 1 × binding buffer. In total, 5 µL of Annexin V-FITC and 5 µL of PI staining solution were added to the sample, and the mixture was incubated in a dark environment for 10 min. Then, 400 µL of 1 × binding buffer was added, and the sample was gently mixed. Finally, the rates of cell apoptosis were measured using a flow cytometer.

### 4.6. Measurement of the Levels of Intracellular ROS

A total of 2 × 10^5^ cells/well were seeded in 6-well plates. A Reactive Oxygen Species Assay Kit (Beyotime) was used to detect the ROS levels. First, we removed the medium and washed the cells with PBS. Then, we added 1 mL of DCFH-DA at a concentration of 10 µmol/L and incubated the mixture for 20 min in a 37 °C cell incubator. After the detachment and separation of the cells using trypsin for 3 min, the samples were centrifuged to collect the cells. Finally, the cells were washed 3 times with PBS, and the ROS levels were measured using a flow cytometer.

### 4.7. Cell Viability Assay

A total of 10^4^ cells/well were seeded in 96-well plates, grown for 12 h and then treated with inactivated *E. coli* and *S. aureus*. Next, 10 µL of CCK-8 (Dojindo, Shanghai, China) was added to each well, and the plate was incubated for 24 h. The absorbance at 450 nm was read.

### 4.8. Establishment of LncRNA Library

We sent 9 samples that underwent different treatments to Cloud-Seq Biotech (Shanghai, China) for lncRNA sequencing. A NEBNext^®^ rRNA Depletion Kit (New England Biolabs, Ipswich USA) was used to enrich for mRNAs by removing rRNAs. To construct RNA libraries with the TruSeq Stranded Total RNA Library Prep Kit (Illumina, San Diego, CA, USA), rRNA-depleted RNAs were used. Finally, the libraries were checked for quality and quantified using the BioAnalyzer 2100 system (Agilent Technologies, Santa Clara, CA, USA). Library sequencing was performed on an Illumina NovaSeq instrument with 150 bp paired-end reads.

### 4.9. LncRNA Raw Data Processing and Identification

Reads were aligned to the cow reference genome (ARS-UCD1.2) using the Hisat2 software (v2.0.4). Then, we used the HTSeq software (v0.9.1) to obtain genes, transcripts of length ≥ 200 bp and exons with counts ≥2. Next, the raw counts were normalized using edgeR (v3.16.5), and the differential expression of lncRNAs between the two sets of samples was calculated. Fold change = 2.0, namely, log2(FC) = 1.0, *p* ≤ 0.05, was set as the threshold for screening the differences.

### 4.10. Analysis of Biological Information and Data Processing

We selected the differential expression analyses of the lncRNAs. Analyses of gene ontology (GO) enrichment for the lncRNA target genes were completed using the DAVID software, and *p* < 0.05 was considered significant. We used the DAVID [60] software to test the statistical enrichment of differentially expressed lncRNA target genes in the Kyoto Encyclopedia of Genes and Genomes (KEGG; https://www.kegg.jp/kegg/, accessed on 27 April 2020). Additionally, we used RNAhybrid (2.1.2) to predict lncRNA and miRNA interactions [61]. The downstream mRNAs were predicted using Targetscan (http://www.targetscan.org/vert_72/, accesed on 20 April 2021) for predicted miRNAs.

### 4.11. Statistical Analysis

Data analyses were performed using Flowjo_V10 and Prism 7 (GraphPad software). The results were obtained from three independent experiments, shown as the mean values (±SEMs) of three independent experiments. Student’s *t*-test was used to differentiate groups, and *p* < 0.05 was considered statistically significant.

## 5. Conclusions

Bovine mastitis is a common disease and its exact pathological mechanism is still under study. LncRNAs differentially expressed in mastitis may regulate the expression of target genes through a variety of potential mechanisms, including cis-acting factors and trans-acting factors ceRNAs, which ultimately lead to the occurrence of mastitis. The current study’s findings indicate that differentially expressed lncRNAs were associated with cell injuries in mastitis models induced by *E. coli* and *S. aureus*. Moreover, our study provides a novel insight into the pathogenesis of mastitis via the lncRNA expression profile, which may represent a new target for controlling and preventing mastitis.

## Figures and Tables

**Figure 1 ijms-22-09719-f001:**
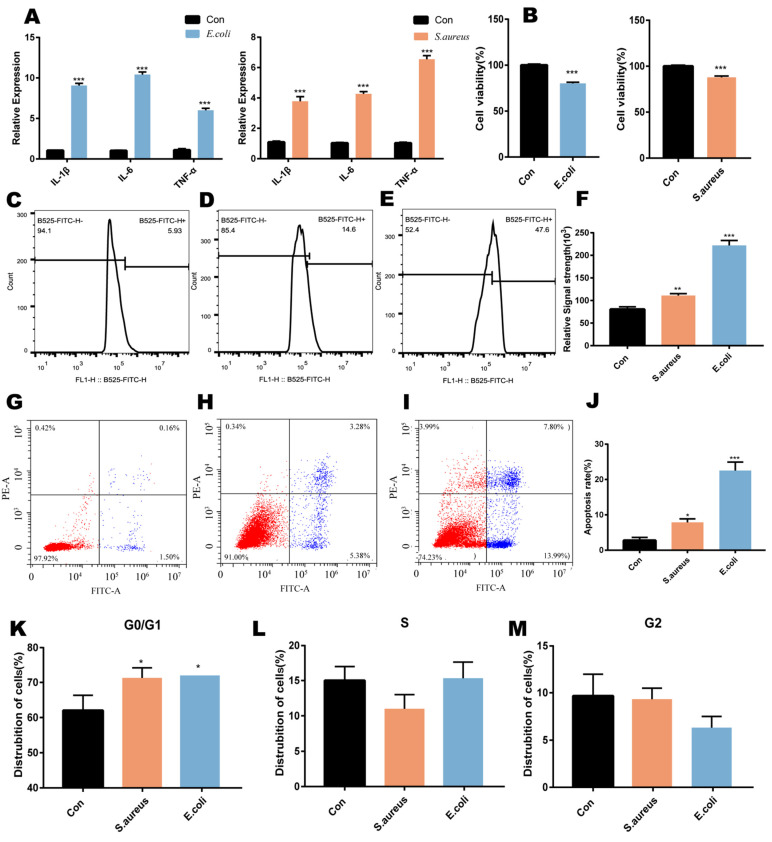
(**A**) mRNA expression of IL-1β, IL-6 and TNF-α. (**B**) Effects of *E. coli* and *S. aureus* on cell viability of MAC-T cells. Detection of ROS levels in cells by flow cytometry: (**C**) Con, (**D**) *E. coli* and (**E**) *S. aureus*. (**F**) Y-axis: signal strength units (10^3^). Apoptosis detected by flow cytometry: viable cells (FITC−/PI−), early apoptotic cells (FITC +/PI-), late apoptotic cells (FITC+/PI+) and cell debris (FITC- /PI +). (**G**) Con, (**I**) *E. coli*, (**H**) *S. aureus*, and (**J**) apoptotic rate. (**K**–**M**) Detection of cycle in cells by flow cytometry. Values are means ± SD, *n* = 3. *, ** and *** represent *p* < 0.05, *p* < 0.01 and *p* < 0.001, respectively.

**Figure 2 ijms-22-09719-f002:**
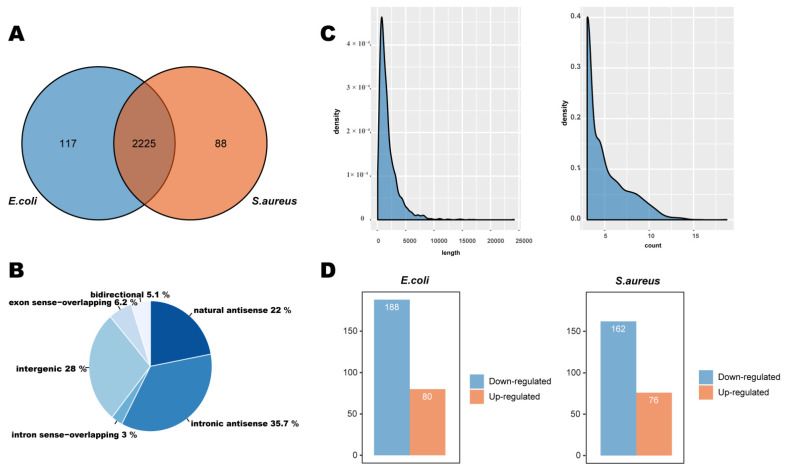
(**A**) Venn diagram showing the numbers of lncRNAs found in the *E. coli* and *S. aureus* groups. (**B**) Novel lncRNAs, mainly classified as intergenic lncRNAs, natural antisense lncRNAs and intronic antisense lncRNAs. (**C**) Distribution of transcript lengths and counts in the lncRNAs. The X-axis depicts the lncRNA length and count, and the Y-axis represents abundance. (**D**) Histogram of the differentially expressed lncRNAs in the *E. coli* and *S. aureus* groups. Columns indicate significantly upregulated lncRNAs (red) and downregulated lncRNAs (blue) (*p* < 0.05).

**Figure 3 ijms-22-09719-f003:**
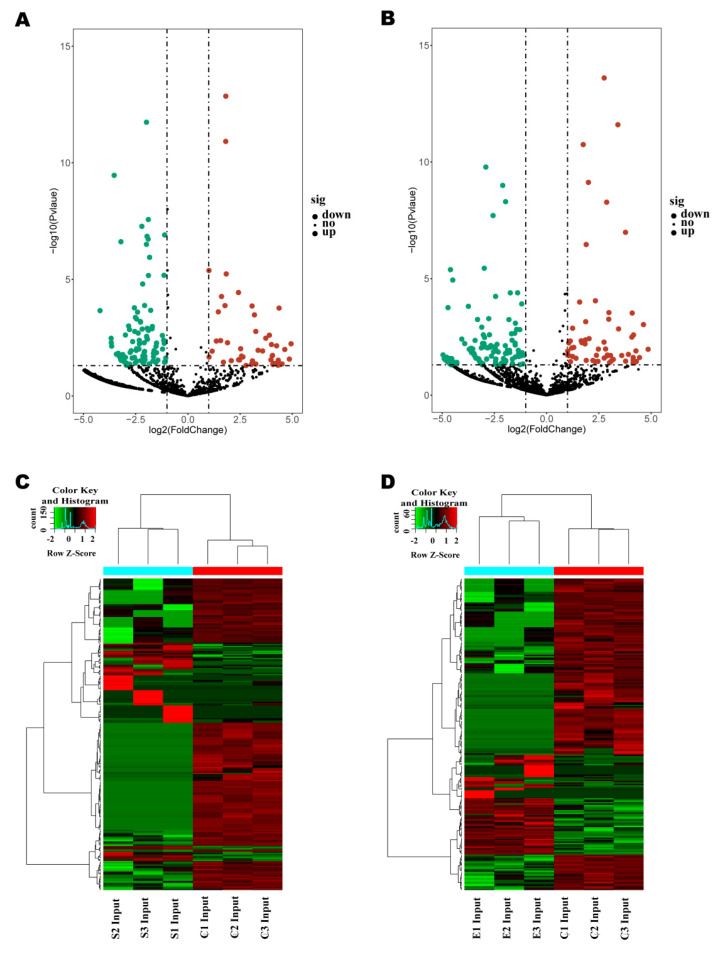
Volcano plots of lncRNAs differentially expressed between the control group, the *E. coli* group (**A**) and the *S. aureus* group (**B**). The red dots on the right indicate the significantly upregulated lncRNAs, and the green dots on the left indicate the significantly downregulated genes (*p* < 0.05). The black dots represent no significant difference (*p* > 0.05). Heatmap of lncRNAs differentially expressed between the control group, the *E. coli* group (**C**) and the *S. aureus* group (**D**). Green-to-red scale indicates low-to-high lncRNA expression levels. Differentially expressed lncRNAs were defined based on a threshold fold change >1.5 at *p* < 0.05.

**Figure 4 ijms-22-09719-f004:**
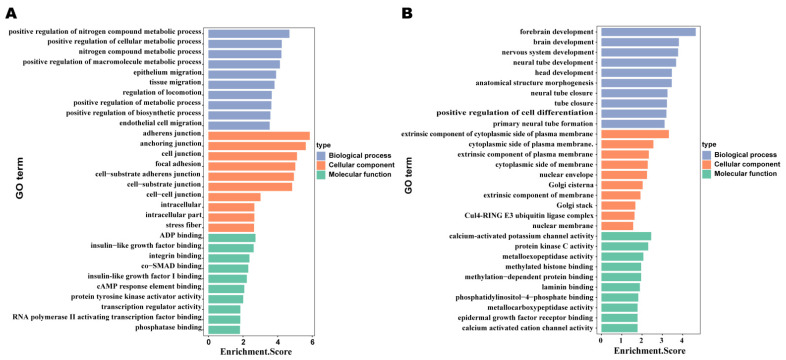
The top 10 GO terms enriched for the targets of significantly downregulated expressed lncRNAs in three categories (biological process, cellular component and molecular function). (**A**) *S. aureus*; (**B**) *E. coli*. *p* ≤ 0.05 = significant enrichment.

**Figure 5 ijms-22-09719-f005:**
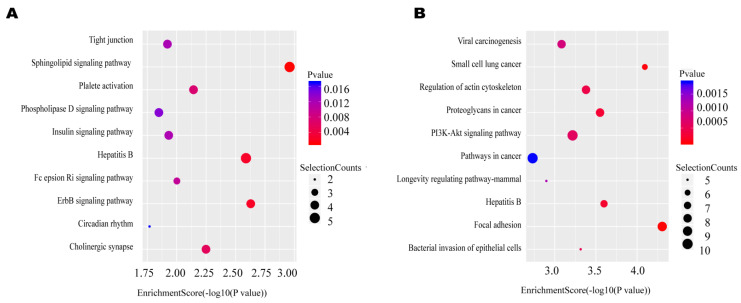
The top 10 KEGG pathways enriched for the targets of significantly downregulated expressed lncRNAs. (**A**) *E. coli*; (**B**) *S. aureus*. *p* ≤ 0.05 = significant enrichment.

**Figure 6 ijms-22-09719-f006:**
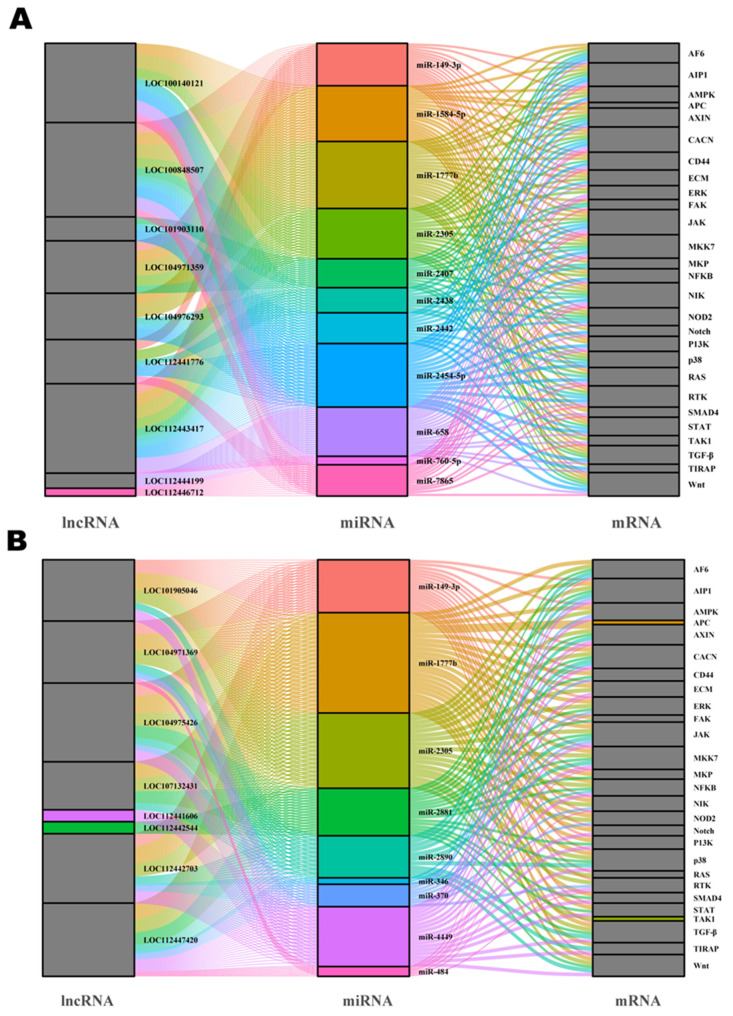
LncRNA–miRNA–mRNA correlation network. The network contains lncRNAs (left) and miRNAs (middle). The size of the square represents how strongly the predicted binding site is associated. The color indicates which miRNA has a predicted binding site for a lncRNA and mRNA. If the predicted number of miRNAs binding is greater than 2, the lncRNA or mRNA is depicted in gray. (**A**) the *E. coli* group, (**B**) the *S. aureus* group.

**Figure 7 ijms-22-09719-f007:**
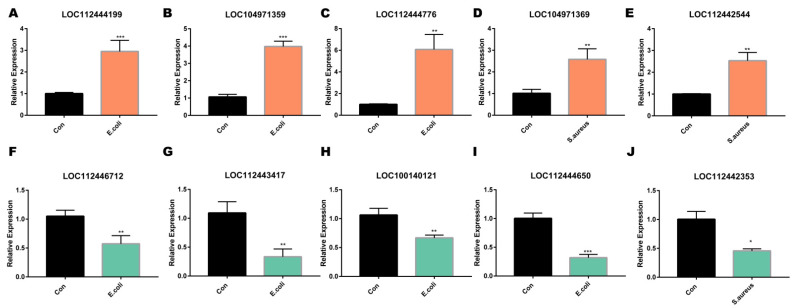
Validation of the differentially expressed lncRNAs by RT-qPCR. (**A**–**E**) the upregulated expressed lncRNAs, (**F**–**J**) the downregulated expressed lncRNAs. Values are means ± SD, *n* = 3. *, ** and *** represent *p* < 0.05, *p* < 0.01 and *p* < 0.001, respectively.

**Figure 8 ijms-22-09719-f008:**
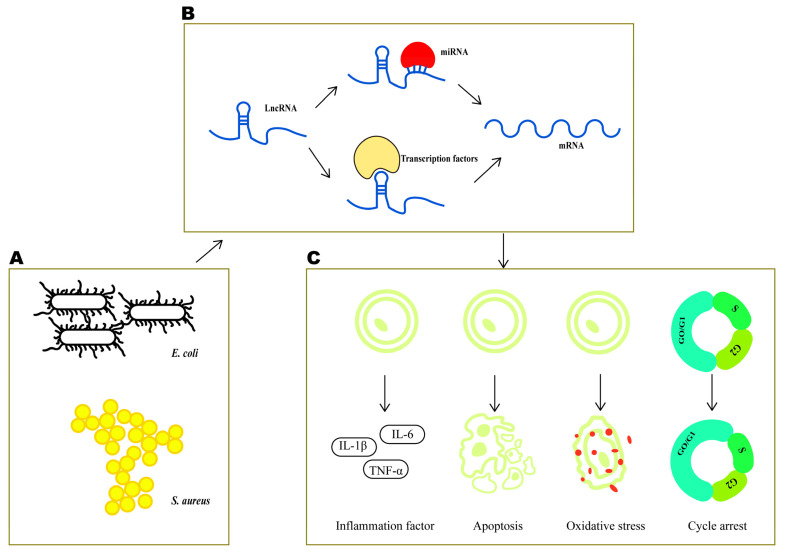
Schematic of relationship between lncRNAs and cell injuries induced by *E. coli* and *S. aureus.* (**A**) Cell injuries inducer of *S. aureus* and *E. coli*, (**B**) LncRNA interact with mRNA in both cis and sponge manners during cell injuries, (**C**) Cell injuries regulated by LncRNA.

**Table 1 ijms-22-09719-t001:** The primers used in this study.

Gene	Primers Sequence (5′ → 3′)	
β-actin	F: TGCTGTCCCTGTATGCCTCT	R: GGTCTTTACGGATGTCAACG
LOC104971359	F: GCTGACCGCTCCTCTCTAAT	R: GTTCTGGCTGGTTCCTGTCA
LOC104971369	F: ACGTGACAAAGTCTGCCGAT	R: TGGTCGCCTGTCCTGTTATG
LOC112444199	F: AAGCAAGGCAGCTCGAGTT	R: CCCCCAGTCCTCATCAAAGT
LOC112444776	F: AGCTTCTCCCCTGGTTTTCC	R: GCACCTTGTCACTCTCCTGA
LOC112446712	F: AGCAGCTGGATAACATGGCA	R: CTTTGGTTTGGTGCACAGGG
LOC112443417	F: CTGCCAGCAACGTGACATTT	R: CTTTGGTTTGGTGCACAGGG
LOC100140121	F: TTTGCGATGACAGGGAAGCT	R: GCGAAATGTAACGCGGGAAA
LOC112442353	F: TGGCAAGAGTCTGGAATGGG	R: CCTCATGGTCACGATGCTCA
LOC112444650	F: AAGACAGTGGATTCGTGGGG	R: TCTGTAAGAATCCCACCGGC
IL-1β	F: TTCCATATTCCTCTTGGGGTAGA	R: AAATGAACCGAGAAGTGGTGTT
IL-6	F: CAGCAGGTCAGTGTTTGTGG	R: CTGGGTTCAATCAGGCGAT
TNF-α	F: CTTCTCAAGCCTCAAGTAACAAGC	R: CCATGAGGGCATTGGCATAC

## Data Availability

https://www.ncbi.nlm.nih.gov/geo/query/acc.cgi?acc=GSE181464, accessed on 5 August 2021.

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
