# Peer review of "Genome-Wide Analysis of LncRNA in Bovine Mammary Epithelial Cell Injuries Induced by Escherichia Coli and Staphylococcus Aureus"

_ijms, 2021, doi:10.3390/ijms22189719_

Round 1

Reviewer 1 Report

The current research aimed to investigate the effects of bovine mammary epithelial cell injuries induced by treatment with E. coli and S. aureus, and explore the lncRNA profile on cell injuries. In general the manuscript is correctly written and it sounds. The current research is novel, correctly presented and it could be of significant scientific value for scientific community. The abstract is informative and clear, the results are clearly presented, interesting and conclusion is logical. Overall, the study provides potentially valuable data regarding demonstrating the dysregulation of lncRNAs in cells upon E. coli and S. aureus in fections, suggesting that they have the potential to become important diagnostic markers and to provide novel insights into controlling and preventing mastitis. However, I suggest that authors add several recent references in the introduction section in order to improve it.

References which I suggest in the manuscript are following:

BUROVIĆ, J. (2020): Isolation of bovine clinical mastitis bacterial pathogens and their antimicrobial susceptibility in the Zenica region in 2017. Vet. stn. 51, 47-52. (In Croatian).

BENIĆ, M., N. MAĆEŠIĆ, L. CVETNIĆ, B. HABRUN, Ž. CVETNIĆ, R. TURK, D. ĐURIČIĆ, M. LOJKIĆ, V. DOBRANIĆ, H. VALPOTIĆ, J. GRIZELJ, D. GRAČNER, J. GRBAVAC, M. SAMARDŽIJA (2018): Bovine mastitis: a persistent and evolving problem requiring novel approaches for its control - a review. Vet. arhiv 88, 535-557

LAMARI, I., N. MIMOUNE, D. KHELEF (2021): Effect of feed additive supplementation on bovine subclinical mastitis. Vet stn. 52, 445-460.

SAIDI, R., Z. CANTEKIN, N. MIMOUNE, Y. ERGUN, H. SOLMAZ, D. KHELEF and R. KAIDI (2021): Investigation of the presence of slime production, VanA gene and antiseptic resistance genes in Staphylococci isolated from bovine mastitis in Algeria. Vet. stn. 52, 57-63.

CVETNIĆ, L., M. SAMARDŽIJA, S. DUVNJAK, B. HABRUN, M. CVETNIĆ, V. JAKI TKALEC, D. ĐURIČIĆ, M. BENIĆ (2021): Multi Locus Sequence Typing and spa Typing of Staphylococcus aureus Isolated from the Milk of Cows with Subclinical Mastitis in Croatia. Microorganisms 9, 725.

MIMOUNE, N., R. SAIDI, O. BENADJEL, D. KHELEF, R. KAIDI (2021): Alternative treatment of bovine mastitis. Vet. Stn. 52, 639-649.

KNEŽEVIĆ, K., V. DOBRANIĆ, D. ĐURIČIĆ, M. SAMARDŽIJA, M. BENIĆ, I. GETZ, M. EFENDIĆ, L. CVETNIĆ, J. ŠAVORIĆ, I. BUTKOVIĆ, M. CVETNIĆ, M. MAZIĆ, N. MAĆEŠIĆ (2021): Use of somatic cell count in the diagnosis of mastitis and its impacts on milk quality. Vet. stn. 52, 751-764. (In Croatian).

Author Response

Response to Reviewer 1 Comments

The current research aimed to investigate the effects of bovine mammary epithelial cell injuries induced by treatment with E. coli and S. aureus, and explore the lncRNA profile on cell injuries. In general the manuscript is correctly written and it sounds. The current research is novel, correctly presented and it could be of significant scientific value for scientific community. The abstract is informative and clear, the results are clearly presented, interesting and conclusion is logical. Overall, the study provides potentially valuable data regarding demonstrating the dysregulation of lncRNAs in cells upon E. coli and S. aureus infections, suggesting that they have the potential to become important diagnostic markers and to provide novel insights into controlling and preventing mastitis.

Point: However, I suggest that authors add several recent references in the introduction section in order to improve it. References which I suggest in the manuscript are following:

BUROVIĆ, J. (2020): Isolation of bovine clinical mastitis bacterial pathogens and their antimicrobial susceptibility in the Zenica region in 2017. Vet. stn. 51, 47-52. (In Croatian).

BENIĆ, M., N. MAĆEŠIĆ, L. CVETNIĆ, B. HABRUN, Ž. CVETNIĆ, R. TURK, D. ĐURIČIĆ, M. LOJKIĆ, V. DOBRANIĆ, H. VALPOTIĆ, J. GRIZELJ, D. GRAČNER, J. GRBAVAC, M. SAMARDŽIJA (2018): Bovine mastitis: a persistent and evolving problem requiring novel approaches for its control - a review. Vet. arhiv 88, 535-557

LAMARI, I., N. MIMOUNE, D. KHELEF (2021): Effect of feed additive supplementation on bovine subclinical mastitis. Vet stn. 52, 445-460.

SAIDI, R., Z. CANTEKIN, N. MIMOUNE, Y. ERGUN, H. SOLMAZ, D. KHELEF and R. KAIDI (2021): Investigation of the presence of slime production, VanA gene and antiseptic resistance genes in Staphylococci isolated from bovine mastitis in Algeria. Vet. stn. 52, 57-63.

CVETNIĆ, L., M. SAMARDŽIJA, S. DUVNJAK, B. HABRUN, M. CVETNIĆ, V. JAKI TKALEC, D. ĐURIČIĆ, M. BENIĆ (2021): Multi Locus Sequence Typing and spa Typing of Staphylococcus aureus Isolated from the Milk of Cows with Subclinical Mastitis in Croatia. Microorganisms 9, 725.

MIMOUNE, N., R. SAIDI, O. BENADJEL, D. KHELEF, R. KAIDI (2021): Alternative treatment of bovine mastitis. Vet. Stn. 52, 639-649.

KNEŽEVIĆ, K., V. DOBRANIĆ, D. ĐURIČIĆ, M. SAMARDŽIJA, M. BENIĆ, I. GETZ, M. EFENDIĆ, L. CVETNIĆ, J. ŠAVORIĆ, I. BUTKOVIĆ, M. CVETNIĆ, M. MAZIĆ, N. MAĆEŠIĆ (2021): Use of somatic cell count in the diagnosis of mastitis and its impacts on milk quality. Vet. stn. 52, 751-764. (In Croatian).

Response: Thanks for your suggestion.

1) The references "Burović et al…. ", " Saidi et al…. ", " Cvetnic, et al…. ", " Lamari, et al…. "and" Knežević et al…. " were added in the Introduction section (Lines 38-43).

2) The references " Mimoune et al…. " and " Benić et al…. " were added in the Discussion section (Line 225).

Reviewer 2 Report

The study proposed by Lin and colleagues is very interesting and it provides to amplify the knowledge of the E.coli and S. aureus epidemiology which cause mastitis in cows, thank LNCRNA research.

The study is well conducted and I don't have any comment concerning it. However, the English must be revised before pubblication.

Author Response

Response to Reviewer 2 Comments

The study proposed by Lin and colleagues is very interesting and it provides to amplify the knowledge of the E.coli and S. aureus epidemiology which cause mastitis in cows, thank LNCRNA research.

The study is well conducted and I don't have any comment concerning it. However, the English must be revised before publication.

Point 1: However, the English must be revised before publication.

Response: Thanks for your suggestion.

The English language were revised extensively by English editors, and the English Editing Certificate was provided.

Reviewer 3 Report

In this manuscript authors constructed a model of mastitis in bovine mammary epithelial cells induced by E. coli and S. aureus and profiled the expression of long non-coding RNAs. The results showed that the expression patterns of lncRNAs in the two mastitis groups were different and the target genes of lncRNAs were enriched in different signaling pathway, respectively. These results shed light on issues of the function of lncRNA in mastitis diseases, a good foundation for a further exploration in this field, but there are some issues that the authors should address to further validate their conclusions:

L20-21:The font size was in this sentence that consistent with the others in the abstract, please check it.

Figure1: The order of the three groups in F and J was different with that in K, please make them consistent; the color of the columns that represented the E.coli or S.aureus group in A was different with the legend, please correct it.

L88-89: The sentence “We observed a significant increase in the cytokines’ control group compared with E. coli and S. aureus (p <0.001)” was not clear, please check it and make it understandable.

L92: “E. coli had a significant effect on cells, while the damaging effects of S. aureus were not significant.” This describe was not consistent with the results showed in Figure1B, since we observed “***” on the column in S. aureus group, please check it.

L138-142: “By constructing a heat map, we were able to show that the two inactivated-bacterium-treated groups produced two separate clusters, indicating that the expression patterns of lncRNAs in the two inactivated-bacterium-treated groups were different from those for the control group, and that the observed differences were significant (Figure 4).” This sentence described the result of heat map, so here it showed be figure 3, please confirm and make a revise if necessary.

L172: “bacterial invasion of epithelial cells. (Figure 5B, Figure S2B)” deleted the “.”

L209: In the figure legend of Figure 6, please state that which group that “A” and “B” represent, respectively. Also annotate “Figure6A” and “Figure6B” in the results description from line188 to line 200, to make your results much more clear.

Author Response

Response to Reviewer 3 Comments

In this manuscript authors constructed a model of mastitis in bovine mammary epithelial cells induced by E. coli and S. aureus and profiled the expression of long non-coding RNAs. The results showed that the expression patterns of lncRNAs in the two mastitis groups were different and the target genes of lncRNAs were enriched in different signaling pathway, respectively. These results shed light on issues of the function of lncRNA in mastitis diseases, a good foundation for a further exploration in this field, but there are some issues that the authors should address to further validate their conclusions:

Point 1: L20-21: The font size was in this sentence that consistent with the others in the abstract, please check it.
Response: Thanks for your suggestion. We have unified the front size in this sentence (Lines 20-22).

Point 2: Figure1: The order of the three groups in F and J was different with that in K, please make them consistent; the color of the columns that represented the E.coli or S.aureus group in A was different with the legend, please correct it.

Response: Thanks for your suggestion.

1) The order of the three groups in Figure 1F and 1J was revised according to the order of Figure 1K (Figure 1).

2) The legends in Figure 1 were revised according to the colour of the columns (Figure 1).

Point 3: L88-89: The sentence “We observed a significant increase in the cytokines’ control group compared with E. coli and S. aureus (p <0.001)” was not clear, please check it and make it understandable.

Response: Thanks for your suggestion.

We replaced “We observed a significant increase ….” with “We observed that there were significant increases of IL-1β, IL-6 and TNF-ɑ…” (Lines 90-92).

Point 4: L92: “E. coli had a significant effect on cells, while the damaging effects of S. aureus were not significant.” This describe was not consistent with the results showed in Figure1B, since we observed “***” on the column in S. aureus group, please check it.

Response: Thanks for your suggestion.

We replaced ““E. coli had a significant effect on cells, while the damaging effects of S. aureus were not significant.” with “The damaging effects of E. coli as well as S. aureus were significant” (Lines 94-95).

Point 5: L138-142: “By constructing a heat map, we were able to show that the two inactivated-bacterium-treated groups produced two separate clusters, indicating that the expression patterns of lncRNAs in the two inactivated-bacterium-treated groups were different from those for the control group, and that the observed differences were significant (Figure 4).” This sentence described the result of heat map, so here it showed be figure 3, please confirm and make a revise if necessary.

Response: Thanks for your suggestion.

1) The sentences “By constructing a heat map…, and that the observed differences were significant (Figure 4) were changed to “By constructing a heat map…, and that the observed differences were significant (Figure 3C, 3D)” (Line141).

2) The sentence “The differentially expressed … for cluster analysis (Figure 3) was changed to “The differentially expressed … for cluster analysis (Figure 3A, 3B).” (Line 145).

Point 6: L172: “bacterial invasion of epithelial cells. (Figure 5B, Figure S2B)” deleted the “.”

Response: Thanks for your suggestion. We deleted the “.” (Line 176).

Point 7: L209: In the figure legend of Figure 6, please state that which group that “A” and “B” represent, respectively. Also annotate “Figure 6A” and “Figure 6B” in the results description from line188 to line 200, to make your results much more clear.

Response: Thanks for your suggestion.

1) We added “(A) the E. coli group, (B) the S. aureus group.” in the figure legend of Figure 6 (Line 216).

2) We added “(Figure 6A)” and “(Figure 6B)” in the results description (Lines189-190).

Round 2

Reviewer 3 Report

I accept this revised version